# Deep Fragment Embeddings for Bidirectional Image Sentence Mapping

**Andrej Karpathy**     **Armand Joulin**     **Li Fei-Fei**
Department of Computer Science, Stanford University, Stanford, CA 94305, USA
{karpathy,ajoulin,feifeili}@cs.stanford.edu

## Abstract

We introduce a model for bidirectional retrieval of images and sentences through a deep, multi-modal embedding of visual and natural language data. Unlike previous models that directly map images or sentences into a common embedding space, our model works on a finer level and embeds fragments of images (objects) and fragments of sentences (typed dependency tree relations) into a common space. We then introduce a structured max-margin objective that allows our model to explicitly associate these fragments across modalities. Extensive experimental evaluation shows that reasoning on both the global level of images and sentences and the finer level of their respective fragments improves performance on image-sentence retrieval tasks. Additionally, our model provides interpretable predictions for the image-sentence retrieval task since the inferred inter-modal alignment of fragments is explicit.

## 1   Introduction

There is significant value in the ability to associate natural language descriptions with images. Describing the contents of images is useful for automated image captioning and conversely, the ability to retrieve images based on natural language queries has immediate image search applications. In particular, in this work we are interested in training a model on a set of images and their associated natural language descriptions such that we can later rank a fixed set of withheld sentences given an image query, and vice versa.

This task is challenging because it requires detailed understanding of the content of images, sentences and their inter-modal correspondence. Consider an example sentence query, such as *"A dog with a tennis ball is swimming in murky water"* (Figure 1). In order to successfully retrieve a corresponding image, we must accurately identify all entities, attributes and relationships present in the sentence and ground them appropriately to a complex visual scene.

Our primary contribution is in formulating a structured, max-margin objective for a deep neural network that learns to embed both visual and language data into a common, multimodal space. Unlike previous work that embeds images and sentences, our model breaks down and embeds fragments of images (objects) and fragments of sentences (dependency tree relations [1]) in a common embedding space and explicitly reasons about their latent, inter-modal correspondences. Extensive empirical evaluation validates our approach. In particular, we report dramatic improvements over state of the art methods on image-sentence retrieval tasks on Pascal1K [2], Flickr8K [3] and Flickr30K [4] datasets. We make our code publicly available.

## 2   Related Work

**Image Annotation and Image Search.**   There is a growing body of work that associates images and sentences. Some approaches focus on the direction of describing the contents of images, formulated either as a task of mapping images to a fixed set of sentences written by people [5, 6], or as a task of automatically generating novel captions [7, 8, 9, 10, 11, 12]. More closely related to our motivation are methods that allow natural bi-drectional mapping between the two modalities. Socher and Fei-Fei [13] and Hodosh et al. [3] use Kernel Canonical Correlation Analysis to align images and sentences, but their method is not easily scalable since it relies on computing kernels quadratic in

Figure 1: Our model takes a dataset of images and their sentence descriptions and learns to associate their fragments. In images, fragments correspond to object detections and scene context. In sentences, fragments consist of typed dependency tree [1] relations.

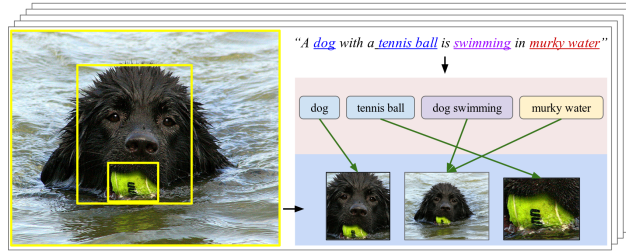

number of images and sentences. Farhadi et al. [5] learn a common meaning space, but their method is limited to representing both images and sentences with a single triplet of (object, action, scene). Zitnick et al. [14] use a Conditional Random Field to reason about spatial relationships in cartoon scenes and their relation to natural language descriptions. Finally, joint models of language and perception have also been explored in robotics settings [15].

**Multimodal Representation Learning.** Our approach falls into a general category of learning from multi-modal data. Several probabilistic models for representing joint multimodal probability distributions over images and sentences have been developed, using Deep Boltzmann Machines [16], log-bilinear models [17], and topic models [18, 19]. Ngiam et al. [20] described an autoencoder that learns audio-video representations through a shared bottleneck layer. More closely related to our task and approach is the work of Frome et al. [21], who introduced a model that learns to map images and words to a common semantic embedding with a ranking cost. Adopting a similar approach, Socher et al. [22] described a Dependency Tree Recursive Neural Network that puts entire sentences into correspondence with visual data. However, these methods reason about the image only on the global level using a single, fixed-sized representation from the top layer of a Convolutional Neural Network as a description for the entire image. In contrast, our model explicitly reasons about objects that make up a complex scene.

**Neural Representations for Images and Natural Language.** Our model is a neural network that is connected to image pixels on one side and raw 1-of-k word representations on the other. There have been multiple approaches for learning neural representations in these data domains. In Computer Vision, Convolutional Neural Networks (CNNs) [23] have recently been shown to learn powerful image representations that support state of the art image classification [24, 25, 26] and object detection [27, 28]. In language domain, several neural network models have been proposed to learn word/n-gram representations [29, 30, 31, 32, 33, 34], sentence representations [35] and paragraph/document representations [36].

## 3 Proposed Model

**Learning and Inference.** Our task is to retrieve relevant images given a sentence query, and conversely, relevant sentences given an image query. We train our model on a set of $N$ images and $N$ corresponding sentences that describe their content (Figure 2). Given this set of correspondences, we learn the weights of a neural network with a structured loss to output a high score when a compatible image-sentence pair is fed through the network, and low score otherwise. Once the training is complete, all training data is discarded and the network is evaluated on a withheld set of images and sentences. The evaluation scores all image-sentence pairs in the test set, sorts the images/sentences in order of decreasing score and records the location of a ground truth result in the list.

**Fragment Embeddings.** Our core insight is that images are complex structures that are made up of multiple entities that the sentences make explicit references to. We capture this intuition directly in our model by breaking down both images and sentences into fragments and reason about their alignment. In particular, we propose to detect objects as image fragments and use sentence dependency tree relations [1] as sentence fragments (Figure 2).

**Objective.** We will compute the representation of both image and sentence fragments with a neural network, and interpret the top layer as high-dimensional vectors embedded in a common multi-modal space. We will think of the inner product between these vectors as a fragment compatibility score, and compute the global image-sentence score as a fixed function of the scores of their respective fragments. Intuitively, an image-sentence pair will obtain a high global score if the sentence fragments can each be confidently matched to some fragment in the image. Finally, we will learn the weights of the neural networks such that the true image-sentence pairs achieve a score higher (by a margin) than false image-sentence pairs.

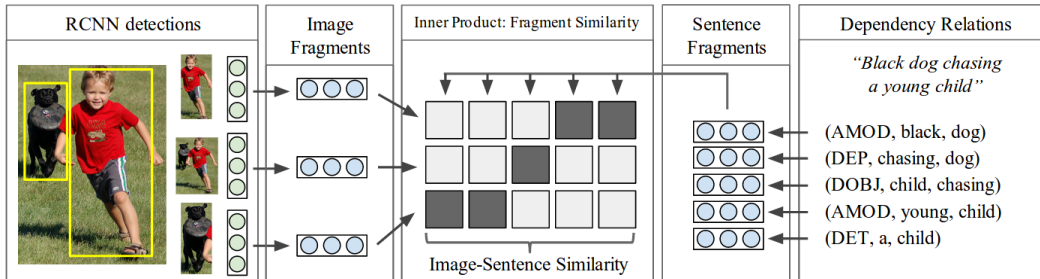

Figure 2: Computing the Fragment and image-sentence similarities. **Left:** CNN representations (green) of detected objects are mapped to the fragment embedding space (blue, Section 3.2). **Right:** Dependency tree relations in the sentence are embedded (Section 3.1). Our model interprets inner products (shown as boxes) between fragments as a similarity score. The alignment (shaded boxes) is latent and inferred by our model (Section 3.3.1). The image-sentence similarity is computed as a fixed function of the pairwise fragment scores.

We first describe the neural networks that compute the Image and Sentence Fragment embeddings. Then we discuss the objective function, which is composed of the two aforementioned objectives.

## 3.1 Dependency Tree Relations as Sentence Fragments

We would like to extract and represent the set of visually identifiable entities described in a sentence. For instance, using the example in Figure 2, we would like to identify the entities (dog, child) and characterise their attributes (black, young) and their pairwise interactions (chasing). Inspired by previous work [5, 22] we observe that a dependency tree of a sentence provides a rich set of typed relationships that can serve this purpose more effectively than individual words or bigrams. We discard the tree structure in favor of a simpler model and interpret each relation (edge) as an individual sentence fragment (Figure 2, right shows 5 example dependency relations). Thus, we represent every word using 1-of-k encoding vector $\mathbf{w}$ using a dictionary of 400,000 words and map every dependency triplet $(R, \mathbf{w}_1, \mathbf{w}_2)$ into the embedding space as follows:

$$s = f\left(W_R \begin{bmatrix} W_e\mathbf{w}_1 \\ W_e\mathbf{w}_2 \end{bmatrix} + b_R\right). \tag{1}$$

Here, $W_e$ is a $d \times 400,000$ matrix that encodes a 1-of-k vector into a $d$-dimensional word vector representation (we use $d = 200$). We fix $W_e$ to weights obtained through an unsupervised objective described in Huang et al. [34]. Note that every relation $R$ can have its own set of weights $W_R$ and biases $b_R$. We fix the element-wise nonlinearity $f(.)$ to be the Rectified Linear Unit (ReLU), which computes $x \rightarrow max(0, x)$. The size of the embedded space is cross-validated, and we found that values of approximately 1000 generally work well.

## 3.2 Object Detections as Image Fragments

Similar to sentences, we wish to extract and describe the set of entities that images are composed of. Inspired by prior work [7], as a modeling assumption we observe that the subject of most sentence descriptions are attributes of objects and their context in a scene. This naturally motivates the use of objects and the global context as the fragments of an image. In particular, we follow Girshick et al. [27] and detect objects in every image with a Region Convolutional Neural Network (RCNN). The CNN is pre-trained on ImageNet [37] and finetuned on the 200 classes of the ImageNet Detection Challenge [38]. We use the top 19 detected locations and the entire image as the image fragments and compute the embedding vectors based on the pixels $I_b$ inside each bounding box as follows:

$$v = W_m[CNN_{\theta_c}(I_b)] + b_m, \tag{2}$$

where $CNN(I_b)$ takes the image inside a given bounding box and returns the 4096-dimensional activations of the fully connected layer immediately before the classifier. The CNN architecture is identical to the one described in Girhsick et al. [27]. It contains approximately 60 million parameters $\theta_c$ and closely resembles the architecture of Krizhevsky et al [25].

## 3.3 Objective Function

We are now ready to formulate the objective function. Recall that we are given a training set of $N$ images and corresponding sentences. In the previous sections we described parameterized functions that map every sentence and image to a set of fragment vectors $\{s\}$ and $\{v\}$, respectively. All parameters of our model are contained in these two functions. As shown in Figure 2, our model

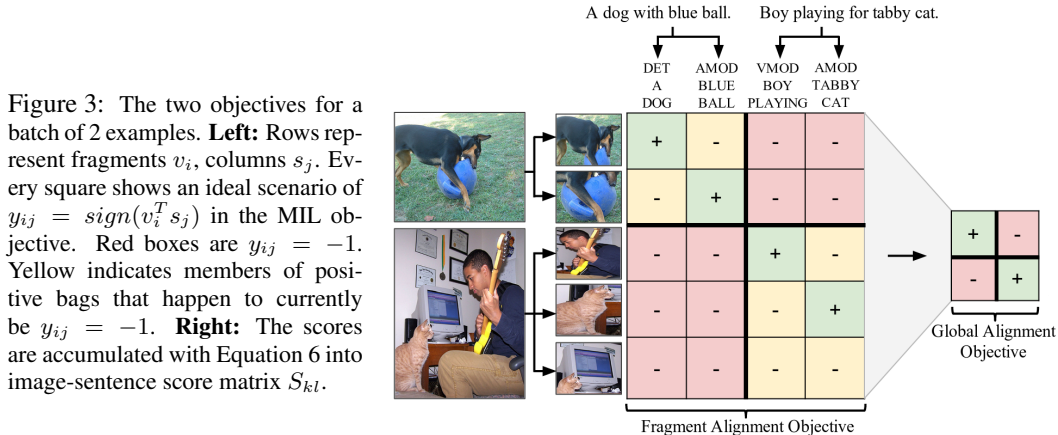

Figure 3: The two objectives for a batch of 2 examples. **Left:** Rows represent fragments $v_i$, columns $s_j$. Every square shows an ideal scenario of $y_{ij} = sign(v_i^T s_j)$ in the MIL objective. Red boxes are $y_{ij} = -1$. Yellow indicates members of positive bags that happen to currently be $y_{ij} = -1$. **Right:** The scores are accumulated with Equation 6 into image-sentence score matrix $S_{kl}$.

then interprets the inner product $v_i^T s_j$ between an image fragment $v_i$ and a sentence fragment $s_j$ as a similarity score, and computes the image-sentence similarity as a fixed function of the scores of their respective fragments.

We are motivated by two criteria in designing the objective function. First, the image-sentence similarities should be consistent with the ground truth correspondences. That is, corresponding image-sentence pairs should have a higher score than all other image-sentence pairs. This will be enforced by the **Global Ranking Objective**. Second, we introduce a **Fragment Alignment Objective** that explicitly learns the appearance of sentence fragments (such as "black dog") in the visual domain. Our full objective is the sum of these two objectives and a regularization term:

$$\mathcal{C}(\theta) = \mathcal{C}_F(\theta) + \beta\mathcal{C}_G(\theta) + \alpha||\theta||_2^2, \tag{3}$$

where $\theta$ is a shorthand for parameters of our neural network ($\theta = \{W_e, W_R, b_R, W_m, b_m, \theta_c\}$) and $\alpha$ and $\beta$ are hyperparameters that we cross-validate. We now describe both objectives in more detail.

### 3.3.1 Fragment Alignment Objective

The Fragment Alignment Objective encodes the intuition that if a sentence contains a fragment (e.g."blue ball", Figure 3), at least one of the boxes in the corresponding image should have a high score with this fragment, while all the other boxes in all the other images that have no mention of "blue ball" should have a low score. This assumption can be violated in multiple ways: a triplet may not refer to anything visually identifiable in the image. The box that the triplet refers to may not be detected by the RCNN. Lastly, other images may contain the described visual concept but its mention may omitted in the associated sentence descriptions. Nonetheless, the assumption is still satisfied in many cases and can be used to formulate a cost function. Consider an (incomplete) Fragment Alignment Objective that assumes a dense alignment between every corresponding image and sentence fragments:

$$\mathcal{C}_0(\theta) = \sum_i \sum_j max(0, 1 - y_{ij}v_i^T s_j). \tag{4}$$

Here, the sum is over all pairs of image and sentence fragments in the training set. The quantity $v_i^T s_j$ can be interpreted as the alignment score of visual fragment $v_i$ and sentence fragment $s_j$. In this incomplete objective, we define $y_{ij}$ as $+1$ if fragments $v_i$ and $s_j$ occur together in a corresponding image-sentence pair, and $-1$ otherwise. Intuitively, $\mathcal{C}_0(\theta)$ encourages scores in red regions of Figure 3 to be less than -1 and scores along the block diagonal (green and yellow) to be more than +1.

**Multiple Instance Learning extension.** The problem with the objective $\mathcal{C}_0(\theta)$ is that it assumes dense alignment between all pairs of fragments in every corresponding image-sentence pair. However, this is hardly ever the case. For example, in Figure 3, the "boy playing" triplet refers to only one of the three detections. We now describe a Multiple Instance Learning (MIL) [39] extension of the objective $\mathcal{C}_0$ that attempts to infer the latent alignment between fragments in corresponding image-sentence pairs. Concretely, for every triplet we put image fragments in the associated image into a positive bag, while image fragments in every other image become negative examples. Our precise formulation is inspired by the *mi-SVM* [40], which is a simple and natural extension of a Support Vector Machine to a Multiple Instance Learning setting. Instead of treating the $y_{ij}$ as constants, we minimize over them by wrapping Equation 4 as follows:

$$\mathcal{C}_F(\theta) = \min_{y_{ij}} \mathcal{C}_0(\theta)$$

$$\text{s.t.} \quad \sum_{i \in p_j} \frac{y_{ij} + 1}{2} \geq 1 \quad \forall j \tag{5}$$

$$y_{ij} = -1 \quad \forall i, j \quad \text{s.t.} \quad m_v(i) \neq m_s(j) \text{ and } y_{ij} \in \{-1, 1\}$$

Here, we define $p_j$ to be the set of image fragments in the positive bag for sentence fragment $j$. $m_v(i)$ and $m_s(j)$ return the index of the image and sentence (an element of $\{1, \ldots, N\}$) that the fragments $v_i$ and $s_j$ belong to. Note that the inequality simply states that at least one of the $y_{ij}$ should be positive for every sentence fragment $j$ (i.e. at least one green box in every column in Figure 3). This objective cannot be solved efficiently [40] but a commonly used heuristic is to set $y_{ij} = sign(v_i^T s_j)$. If the constraint is not satisfied for any positive bag (i.e. all scores were below zero), the highest-scoring item in the positive bag is set to have a positive label.

### 3.3.2 Global Ranking Objective

Recall that the Global Ranking Objective ensures that the computed image-sentence similarities are consistent with the ground truth annotation. First, we define the image-sentence alignment score to be the average thresholded score of their pairwise fragment scores:

$$S_{kl} = \frac{1}{|g_k|(|g_l| + n)} \sum_{i \in g_k} \sum_{j \in g_l} max(0, v_i^T s_j). \tag{6}$$

Here, $g_k$ is the set of image fragments in image $k$ and $g_l$ is the set of sentence fragments in sentence $l$. Both $k, l$ range from $1, \ldots, N$. We truncate scores at zero because in the *mi-SVM* objective, scores greater than 0 are considered correct alignments and scores less than 0 are considered to be incorrect alignments (i.e. false members of a positive bag). In practice, we found that it was helpful to add a smoothing term $n$, since short sentences can otherwise have an advantage (we found that $n = 5$ works well and that this setting is not very sensitive). The Global Ranking Objective then becomes:

$$\mathcal{C}_G(\theta) = \sum_k \Big[ \underbrace{\sum_l max(0, S_{kl} - S_{kk} + \Delta)}_{\text{rank images}} + \underbrace{\sum_l max(0, S_{lk} - S_{kk} + \Delta)}_{\text{rank sentences}} \Big]. \tag{7}$$

Here, $\Delta$ is a hyperparameter that we cross-validate. The objective stipulates that the score for true image-sentence pairs $S_{kk}$ should be higher than $S_{kl}$ or $S_{lk}$ for any $l \neq k$ by at least a margin of $\Delta$.

### 3.4 Optimization

We use Stochastic Gradient Descent (SGD) with mini-batches of 100, momentum of 0.9 and make 20 epochs through the training data. The learning rate is cross-validated and annealed by a fraction of $\times 0.1$ for the last two epochs. Since both Multiple Instance Learning and CNN finetuning benefit from a good initialization, we run the first 10 epochs with the fragment alignment objective $\mathcal{C}_0$ and CNN weights $\theta_c$ fixed. After 10 epochs, we switch to the full MIL objective $\mathcal{C}_F$ and begin finetuning the CNN. The word embedding matrix $W_e$ is kept fixed due to overfitting concerns. Our implementation runs at approximately 1 second per batch on a standard CPU workstation.

## 4 Experiments

**Datasets.** We evaluate our image-sentence retrieval performance on Pascal1K [2], Flickr8K [3] and Flickr30K [4] datasets. The datasets contain 1,000, 8,000 and 30,000 images respectively and each image is annotated using Amazon Mechanical Turk with 5 independent sentences.

**Sentence Data Preprocessing.** We did not explicitly filter, spellcheck or normalize any of the sentences for simplicity. We use the Stanford CoreNLP parser to compute the dependency trees for every sentence. Since there are many possible relation types (as many as hundreds), due to overfitting concerns and practical considerations we remove all relation types that occur less than 1% of the time in each dataset. In practice, this reduces the number of relations from 136 to 16 in Pascal1K, 170 to 17 in Flickr8K, and from 212 to 21 in Flickr30K. Additionally, all words that are not found in our dictionary of 400,000 words [34] are discarded.

**Image Data Preprocessing.** We use the Caffe [41] implementation of the ImageNet Detection RCNN model [27] to detect objects in all images. On our machine with a Tesla K40 GPU, the RCNN processes one image in approximately 25 seconds. We discard the predictions for 200 ImageNet detection classes and only keep the 4096-D activations of the fully connect layer immediately before the classifier at all of the top 19 detected locations and from the entire image.

| Pascal1K | | | | | | | | |
|---|---|---|---|---|---|---|---|---|
| | Image Annotation | | | | Image Search | | | |
| **Model** | **R@1** | **R@5** | **R@10** | **Mean** $r$ | **R@1** | **R@5** | **R@10** | **Mean** $r$ |
| Random Ranking | 4.0 | 9.0 | 12.0 | 71.0 | 1.6 | 5.2 | 10.6 | 50.0 |
| Socher et al. [22] | 23.0 | 45.0 | 63.0 | 16.9 | 16.4 | 46.6 | 65.6 | 12.5 |
| kCCA. [22] | 21.0 | 47.0 | 61.0 | 18.0 | 16.4 | 41.4 | 58.0 | 15.9 |
| DeViSE [21] | 17.0 | 57.0 | 68.0 | 11.9 | 21.6 | 54.6 | 72.4 | 9.5 |
| SDT-RNN [22] | 25.0 | 56.0 | 70.0 | 13.4 | **25.4** | **65.2** | **84.4** | **7.0** |
| Our model | **39.0** | **68.0** | **79.0** | **10.5** | 23.6 | 65.2 | 79.8 | 7.6 |

Table 1: Pascal1K ranking experiments. **R@K** is Recall@K (high is good). **Mean r** is the mean rank (low is good). Note that the test set only consists of 100 images.

| Flickr8K | | | | | | | | |
|---|---|---|---|---|---|---|---|---|
| | Image Annotation | | | | Image Search | | | |
| **Model** | **R@1** | **R@5** | **R@10** | **Med** $r$ | **R@1** | **R@5** | **R@10** | **Med** $r$ |
| Random Ranking | 0.1 | 0.6 | 1.1 | 631 | 0.1 | 0.5 | 1.0 | 500 |
| Socher et al. [22] | 4.5 | 18.0 | 28.6 | 32 | 6.1 | 18.5 | 29.0 | 29 |
| DeViSE [21] | 4.8 | 16.5 | 27.3 | 28 | 5.9 | 20.1 | 29.6 | 29 |
| SDT-RNN [22] | 6.0 | 22.7 | 34.0 | 23 | 6.6 | 21.6 | 31.7 | 25 |
| Fragment Alignment Objective | 7.2 | 21.9 | 31.8 | 25 | 5.9 | 20.0 | 30.3 | 26 |
| Global Ranking Objective | 5.8 | 21.8 | 34.8 | 20 | 7.5 | 23.4 | 35.0 | 21 |
| (†) Fragment + Global | 12.5 | 29.4 | 43.8 | 14 | 8.6 | 26.7 | 38.7 | 17 |
| † → Images: Fullframe Only | 5.9 | 19.2 | 27.3 | 34 | 5.2 | 17.6 | 26.5 | 32 |
| † → Sentences: BOW | 9.1 | 25.9 | 40.7 | 17 | 6.9 | 22.4 | 34.0 | 23 |
| † → Sentences: Bigrams | 8.7 | 28.5 | 41.0 | 16 | 8.5 | 25.2 | 37.0 | 20 |
| Our model († + MIL) | **12.6** | **32.9** | **44.0** | **14** | **9.7** | **29.6** | **42.5** | **15** |
| * Hodosh et al. [3] | 8.3 | 21.6 | 30.3 | 34 | 7.6 | 20.7 | 30.1 | 38 |
| * Our model († + MIL) | **9.3** | **24.9** | **37.4** | **21** | **8.8** | **27.9** | **41.3** | **17** |

Table 2: Flickr8K experiments. **R@K** is Recall@K (high is good). **Med r** is the median rank (low is good). The starred evaluation criterion (*) in [3] is slightly different since it discards 4,000 out of 5,000 test sentences.

**Evaluation Protocol for Bidirectional Retrieval.** For Pascal1K we follow Socher et al. [22] and use 800 images for training, 100 for validation and 100 for testing. For Flickr datasets we use 1,000 images for validation, 1,000 for testing and the rest for training (consistent with [3]). We compute the dense image-sentence similarity $S_{kl}$ between every image-sentence pair in the test set and rank images and sentences in order of decreasing score. For both Image Annotation and Image Search, we report the median rank of the closest ground truth result in the list, as well as Recall@K which computes the fraction of times the correct result was found among the top K items. When comparing to Hodosh et al. [3] we closely follow their evaluation protocol and only keep a subset of $N$ sentences out of total $5N$ (we use the first sentence out of every group of 5).

### 4.1 Comparison Methods

**SDT-RNN.** Socher et al. [22] embed a fullframe CNN representation with the sentence representation from a Semantic Dependency Tree Recursive Neural Network (SDT-RNN). Their loss matches our global ranking objective. We requested the source code of Socher et al. [22] and substituted the Flickr8K and Flick30K datasets. To better understand the benefits of using our detection CNN and for a more fair comparison we also train their method with our CNN features. Since we have multiple objects per image, we average representations from all objects with detection confidence above a (cross-validated) threshold. We refer to the exact method of Socher et al. [22] with a single fullframe CNN as "Socher et al", and to their method with our combined CNN features as "SDT-RNN".

**DeViSE.** The DeViSE [21] source code is not publicly available but their approach is a special case of our method with the following modifications: we use the average (L2-normalized) word vectors as a sentence fragment, the average CNN activation of all objects above a detection threshold (as discussed in case of SDT-RNN) as an image fragment and only use the global ranking objective.

### 4.2 Quantitative Evaluation

**Our model outperforms previous methods.** Our full method consistently outperforms previous methods on Flickr8K (Table 2) and Flickr30K (Table 3) datasets. On Pascal1K (Table 1) the SDT-RNN appears to be competitive on Image Search.

**Fragment and Global Objectives are complementary.** As seen in Tables 2 and 3, both objectives perform well independently, but benefit from the combination. Note that the Global Objective performs consistently better, possibly because it directly minimizes the evaluation criterion (ranking

| | **Flickr30K** | | | | | | | |
|---|---|---|---|---|---|---|---|---|
| | Image Annotation | | | | Image Search | | | |
| **Model** | **R@1** | **R@5** | **R@10** | **Med** $r$ | **R@1** | **R@5** | **R@10** | **Med** $r$ |
| Random Ranking | 0.1 | 0.6 | 1.1 | 631 | 0.1 | 0.5 | 1.0 | 500 |
| DeViSE [21] | 4.5 | 18.1 | 29.2 | 26 | 6.7 | 21.9 | 32.7 | 25 |
| SDT-RNN [22] | 9.6 | 29.8 | 41.1 | 16 | 8.9 | 29.8 | 41.1 | 16 |
| Fragment Alignment Objective | 11 | 28.7 | 39.3 | 18 | 7.6 | 23.8 | 34.5 | 22 |
| Global Ranking Objective | 11.5 | 33.2 | 44.9 | 14 | 8.8 | 27.6 | 38.4 | 17 |
| (†) Fragment + Global | 12.0 | 37.1 | 50.0 | 10 | 9.9 | 30.5 | 43.2 | 14 |
| Our model († + MIL) | 14.2 | 37.7 | 51.3 | 10 | 10.2 | 30.8 | 44.2 | 14 |
| Our model + Finetune CNN | **16.4** | **40.2** | **54.7** | **8** | **10.3** | **31.4** | **44.5** | **13** |

Table 3: Flickr30K experiments. **R@K** is Recall@K (high is good). **Med** $r$ is the median rank (low is good).

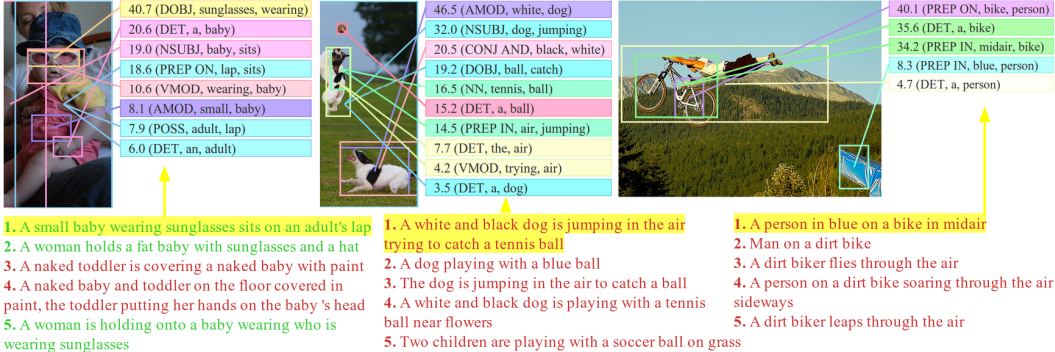

Figure 4: Qualitative Image Annotation results. Below each image we show the top 5 sentences (among a set of 5,000 test sentences) in descending confidence. We also show the triplets for the top sentence and connect each to the detections with the highest compatibility score (indicated by lines). The numbers next to each triplet indicate the matching fragment score. We color a sentence green if it correct and red otherwise.

cost), while the Fragment Alignment Objective only does so indirectly.

**Extracting object representations is important.** Using only the global scene-level CNN representation as a single fragment for every image leads to a consistent drop in performance, suggesting that a single fullframe CNN alone is inadequate in effectively representing the images. (Table 2)

**Dependency tree relations outperform BoW/bigram representations.** We compare to a simpler Bag of Words (BoW) baseline to understand the contribution of dependency relations. In BoW baseline we iterate over words instead of dependency triplets when creating bags of sentence fragments (set $\mathbf{w}_1 = \mathbf{w}_2$ in Equation1). As can be seen in the Table 2, this leads to a consistent drop in performance. This drop could be attributed to a difference between using one word or two words at a time, so we also compare to a bigram baseline where the words $\mathbf{w}_1, \mathbf{w}_2$ in Equation 1 refer to consecutive words in a sentence, not nodes that share an edge in the dependency tree. Again, we observe a consistent performance drop, which suggests that the dependency relations provide useful structure that the neural network takes advantage of.

**Finetuning the CNN helps on Flickr30K.** Our end-to-end Neural Network approach allows us to backpropagate gradients all the way down to raw data (pixels or 1-of-k word encodings). In particular, we observed additional improvements on the Flickr30K dataset (Table 3) when we finetune the CNN. Training the CNN improves the validation error for a while but the model eventually starts to overfit. Thus, we found it critical to keep track of the validation error and freeze the model before it overfits. We were not able to improve the validation performance on Pascal1K and Flickr8K datasets and suspect that there is an insufficient amount of training data.

### 4.3 Qualitative Experiments

**Interpretable Predictions.** We show some example sentence retrieval results in Figure 4. The alignment in our model is explicitly inferred on the fragment level, which allows us to interpret the scores between images and sentences. For instance, in the last image it is apparent that the model retrieved the top sentence because it erroneously associated a mention of a blue person to the blue flag on the bottom right of the image.

**Fragment Alignment Objective trains attribute detectors.** The detection CNN is trained to predict one of 200 ImageNet Detection classes, so it is not clear if the representation is powerful enough to support learning of more complex attributes of the objects or generalize to novel classes. To see whether our model successfully learns to predict sentence triplets, we fix a triplet vector and

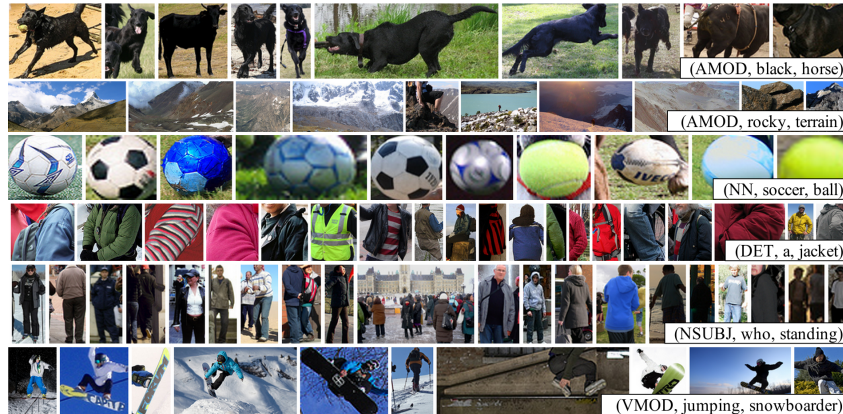

Figure 5: We fix a triplet and retrieve the highest scoring image fragments in the test set. Note that ball, person and dog are ImageNet Detection classes but their visual properties (e.g. soccer, standing, snowboarding, black) are not. Jackets and rocky scenes are not ImageNet Detection classes. Find more in supplementary material.

search for the highest scoring boxes in the test set. Qualitative results shown in Figure 5 suggest that the model is indeed capable of generalizing to more fine-grained subcategories (such as "black dog", "soccer ball") and to out of sample classes such as "rocky terrain" and "jacket".

**Limitations.** Our model is subject to multiple limitations. From a modeling perspective, the use of edges from a dependency tree is simple, but not always appropriate. First, a single complex phrase that describes a single visual entity can be split across multiple sentence fragments. For example, "black and white dog" is parsed as two relations *(CONJ, black, white)* and *(AMOD, white, dog)*. Conversely, there are many dependency relations that don't have a clear grounding in the image (for example "each other"). Furthermore, phrases such as "three children playing" that describe some particular number of visual entries are not modeled. While we have shown that the relations give rise to more powerful representations than words or bigrams, a more careful treatment of sentence fragments will likely lead to further improvements. On the image side, the non-maximum suppression in the RCNN can sometimes detect, for example, multiple people inside one person. Since the model does not take into account any spatial information associated with the detections, it is hard for it to disambiguate between two distinct people or spurious detections of one person.

## 5  Conclusions

We addressed the problem of bidirectional retrieval of images and sentences. Our neural network learns a multi-modal embedding space for fragments of images and sentences and reasons about their latent, inter-modal alignment. We have shown that our model significantly improves the retrieval performance on image sentence retrieval tasks compared to previous work. Our model also produces interpretable predictions. In future work we hope to develop better sentence fragment representations, incorporate spatial reasoning, and move beyond bags of fragments.

**Acknowledgments.** We thank Justin Johnson and Jon Krause for helpful comments and discussions. We gratefully acknowledge the support of NVIDIA Corporation with the donation of the GPUs used for this research. This research is supported by an ONR MURI grant, and NSF ISS-1115313.

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
