[Reviews · NeurIPS 2014]

Submitted by Assigned_Reviewer_32

The paper proposes a method for relating images and sentences by optimizing over the mapping of sentence fragments and image regions. The method uses existing word vector and image region representations. Experiments show that this method is better able to rank human-generated image captions.

Quality
The paper is high quality, and well-supported by empirical results.

Clarity
The paper clearly motivates and describes each of the terms in the objective function.

Originality
Departures from existing work are clear. It may be worth discussing the relation of this paper to work on grounded language learning (e.g., Matuszek et al. ICML 2012).

Significance
The results of this paper are better than previous work and are elucidated by ablation tests.
Summary: The paper is a straightforward extension of existing techniques that leads to great empirical results.

Submitted by Assigned_Reviewer_33

This paper proposes a novel approach to learning joint image-text embeddings for the tasks of image to text retrieval and vice versa. The authors propose the use of image and sentence fragments, which come from convolutional net features of detected objects and dependency tree relations. A ranking objective is proposed that combines both a fragment and global objective, where latent dependency-fragment alignments are inferred with a MIL extension. Extensive experimentation is performed on the ranking tasks proposed by Hodosh et al, including results on the new Flickr30k dataset.

This is a nice paper. The authors obtain strong performance on each of the datasets and the experimentation is well done. Model choices are experimentally verified with convincing results. I have some minor questions / comments below that I hope the authors can respond to:

- lines 018-021: While this work is the first to go beyond a global ranking objective, conceivably even previous methods that work globally should be able to reason locally. For example, each intermediate node of an RNN would live within the same embedding space (as well as words, if they are fine-tuned). This would allow, for example, the retrieval of images from a single word or short phrase, even through training was only done globally. Of course, I would expect better results from your approach since such fragment-level embeddings are being optimized for directly.

- line 137: Wouldn't W_e be d x 400,000, since each of w_1, w_2 is 400,000 x 1?

- It would be helpful to summarize the space of hyperparameters that you tune (there seems to be a lot of them). Do the best validation hyperparameter choices vary greatly across datasets? Any intuition about the hyperparameter space (are some choices consistently better than others, etc) would be helpful for the reader to gain some intuition. How sensitive are the validation results to \beta (the weight on the global objective)?

- On a related note, even though you were not able to cross-validate all the hyperparameters of Flickr30k with SDT-RNN, perhaps you could report results on Flickr30k using the best selected hyperparameters from Flickr8k (unless there is overlap between the two datasets)

- section 4.1 baselines: Are you also including the CNN features for the full image, or just the detected objects?

- You mention that the class predictions from the convolutional net are discarded. Wouldn't these have been useful to inferring the fragment alignments?

- One of the challenges of this task is obtaining large enough datasets with detailed, accurate image captions. The experimental results are convincing in this case, since the three datasets used went through an extensive annotation process. How do you think your method would do on larger, but less accurate captioned datasets? For example, the SBU captioned photo dataset has 1 million images but many of these captions do not necessarily depect visual scenes (e.g. 'fido' referring to a dog). On such datasets, I would suspect that the use of fragment embeddings would be less beneficial.
Summary: This is a good paper with a novel algorithm for image-text retrieval with extensive experimentation and strong results. I recommend acceptance.

Submitted by Assigned_Reviewer_37

This paper proposes a model for bidirectional retrieval of image and sentences using convolutional network features and learned word embeddings. The novelty is the introduction of a fragment level representation and a fragment-level loss function.

The experiments are thorough and demonstrate better performance across the board compared to other the state-of-the-art models.

There are two things this paper uses that other paper did not. One is the object detection results and the other is the new fragment-level loss. It would be nice to isolate the contribution of each. For example, how would [20] do if it could use the fragments (other than simple average). Also how important is the use of dependency relations compared to other types of word embeddings.

It is a little bit surprising that with 30K data in table 3, finetuning the CNN (which is trained using large imagenet) can achieve much better results.
Summary: This is a good paper with novel ideas, supporting experiments and state of the art perfomances.
Author Feedback
Author rebuttal: We thank the reviewers for feedback and appreciate positive comments about the paper's clarity and experiments. We first address the major comments by the reviewers, and then briefly discuss the minor comments.

The impact scores of two reviewers describes our work as incremental.. We would like to note that the idea of translating between natural language and visual data with neural networks trained with a structured, max-margin loss function has been introduced before us. However, to our knowledge, no work has so far tried to decompose both modalities into their respective parts and reason about their explicit (latent) alignment. Our model does so through a novel network architecture and objective. As a result, unlike any previous work, our model produces interpretable groundings (e.g. the sentence fragment “black dog” is explicitly aligned to some particular region of the image) and our experiments show that this formulation also leads to significantly better performance than prior work, across three different datasets.

Reviewer 32:
- The work of Matuszek et al. falls into a category of grounded learning in robotics, which we can include in our Related Work. Compared to this work, our task is primarily focused on ranking rather than attribute classification, our visual domain consists of more uncontrolled web images (without depth), and our training data is more weakly labeled since we do not assume explicit grounding annotations in the training data.

Reviewer 33:
- We omitted the SDT-RNN results on Flickr30K because the long training times (on orders of several days) prevented us from satisfyingly cross-validating the parameters. The reviewer suggests that we use the Flickr8K final SDT-RNN hyperparameters to report our results for Flickr30K. Unfortunately this is not straight-forward because Flickr8K and Flickr30K overlap. However, Table 2 shows that the SDT-RNN is not competitive with our method on Flickr8K, achieving only approximately 77% of our performance. Despite these issues, we can include our best SDT-RNN result on Flickr30K, with a note in text that mentions these concerns.

- The reviewer correctly observes that we could, in principle, make use of the softmax weights form the RCNN model, for example to initialize our image fragment mapping W_m in equation 2. However, this is not straight-forward, since the softmax prediction is 200-dimensional (line 148) but our fragment embeddings vary between 500-1000 dimensions. Moreover, our dataset contains many visual concepts that are not an ImageNet detection class (e.g. mountain, snowboarding, Figure 5), and conversely several ImageNet detection classes occur infrequently in our data (e.g. lobster, pencil sharpener, giant panda, etc.).

- As the reviewer points out, there is a tradeoff between the size of dataset and the quality of associated captions. In this work we focused on relatively small datasets with good annotations (sentence descriptions from AMT rather than image captions). We consider this to be a first step and believe that there is more work to be done in understanding how these methods scale to larger and more weakly-labeled datasets.

Reviewer 37:
- We make use of the fragment representations to try to establish a fair comparison with the model of Socher et al. [20] and Frome et al. [19] (line 311). Since their models assume a single vector describing the image, we took an average of the vector for each object in the image as the most natural extension to our setting. This choice is motivated by Natural Language Processing literature, where it is common to average word vectors to obtain baseline sentence-level representations (For example, as done in Socher et al [20]). It is not immediately obvious how one might apply an RNN [20] on image fragments, because unlike sentences where one can extract a Dependency Tree or a Parse Tree, there is no clear way of imposing a tree structure on image detections.

- As the reviewer observes, there is an insufficient amount of data in Flickr30K to fully train a 60-million parameter CNN without overfitting. As with all other experiments, we keep track of the validation performance (as described on line 298) to pick the best CNN checkpoint as we fine-tune the net. We found that the validation performance improves initially when we begin to finetune the RCNN network, but after a while the model starts to overfit. We will make this more explicit in text.

Minor comments:
Reviewer 33:
- Line 137 is a typo, as indicated. Thank you for pointing this out.
- A as correctly noted, our method involves a few hyper-parameters that are cross-validated. We mention a few of these settings (e.g. lines 240, 251), and as mentioned at the end of Introduction, we also plan on releasing our learning and cross-validation code to further support this discussion.
- Section 4.1 baseline: As mentioned on line 149, our image fragments always include the full-frame features in addition to the top 19 detections.